# Carbohydrases and Phytase in Poultry and Pig Nutrition: A Review beyond the Nutrients and Energy Matrix

**DOI:** 10.3390/ani14020226

**Published:** 2024-01-11

**Authors:** Dante Teixeira Valente Junior, Jansller Luiz Genova, Sung Woo Kim, Alysson Saraiva, Gabriel Cipriano Rocha

**Affiliations:** 1Muscle Biology and Nutrigenomics Laboratory, Department of Animal Sciences, Universidade Federal de Viçosa, Viçosa 36570-900, MG, Brazil; dante.junior@ufv.br (D.T.V.J.); jansller.genova@ufv.br (J.L.G.); alysson.saraiva@ufv.br (A.S.); 2Department of Animal Science, North Carolina State University, Raleigh, NC 27695, USA; sungwoo_kim@ncsu.edu

**Keywords:** carbohydrases, nutritional matrix, phytase, pigs, poultry, sanitary challenge

## Abstract

**Simple Summary:**

Cereal grains and their by-products incorporated into non-ruminant animal feeds contain non-starch polysaccharides and phytate, which are anti-nutritional factors. The animals’ endogenous digestive enzymes do not break down non-starch polysaccharides and phytate. Thus, exogenous enzymes such as carbohydrases and phytase, are supplemented to increase dietary energy and nutrient availability, reducing dietary cost and improving growth performance. Nevertheless, recent studies reveal that the improvement in growth performance goes beyond the release of indigestible nutrients. The positive impacts extend to the intestinal microbiota, immune system, and antioxidant status. However, it is unclear whether the benefits of enzyme action can be translated into better animal growth on commercial farms that face environmental, immunological, and management challenges. In this review, we observed that the supplementation of carbohydrases and phytase under poor sanitary conditions aligns with the energy and nutritional valorization matrices. This suggests that enzymes have properties that promote overall intestinal health, because they expedite post-sanitary challenge recovery and, at the same time, maintain improved utilization of the nutritional matrix. Additionally, studies under commercial conditions demonstrate that matrices containing carbohydrases and phytase sustain the growth and general health of broiler chickens and pigs. However, future research is needed to determine the extent of energy and nutrient savings at the commercial farm level.

**Abstract:**

This review aimed to clarify the mechanisms through which exogenous enzymes (carbohydrases and phytase) influence intestinal health, as well as their effects on the nutrients and energy matrix in diets fed to poultry and pigs reared under sanitary challenging conditions. Enzyme supplementation can positively affect intestinal microbiota, immune system, and enhance antioxidant status. Although enzymes have been shown to save energy and nutrients, their responses under sanitary challenging conditions are poorly documented. Immune system activation alters nutrient partitioning, which can affect the matrix values for exogenous enzymes on commercial farms. Notably, the carbohydrases and phytase supplementation under sanitary challenging conditions align with energy and nutritional valorization matrices. Studies conducted under commercial conditions have shown that matrices containing carbohydrases and phytase can maintain growth performance and health in poultry and pigs. However, these studies have predominantly focused on assessing a single level of reduction in energy and/or available phosphorus and total calcium, limiting our ability to quantify potential energy and nutrient savings in the diet. Future research should delve deeper into determining the extent of energy and nutrient savings and understanding the effects of alone or blended enzymes supplementation to achieve more specific insights.

## 1. Introduction

Cereal grains and co-products used in feeds for non-ruminant animals contain non-starch polysaccharides (NSP) fractions such as arabinoxylans, cellulose, and β-glucans, which are structurally linked with the cell wall and are not broken down by the endogenous digestive enzymes from the animals [1,2]. Furthermore, phytate, an antinutritional compound present in cereal grains and co-products, is only partially accessible to non-ruminant animals due to the limited endogenous phytase activity in their small intestine [3]. Consequently, phytate diminishes the bioavailability of essential minerals (e.g., P, Ca, Zn, and Na), amino acids, and carbohydrates [3,4]. As a result, the inefficiency in digesting these antinutritional compounds has led to the development and application of exogenous feed enzyme technology [5,6].

Historically, application of carbohydrases and phytase has focused on enhancing the nutritional matrix of energy and P [3,5,7]. Interestingly, regardless of the lack of changes in nutrient digestibility, non-ruminant animals fed exogenous enzyme complexes showed better overall performance because exogenous enzymes provide other benefits for intestinal health [8,9,10]. Additionally, studies have reported improvements in growth performance resulting from the supplementation of exogenous enzymes, even in animals fed diets containing limited available substrates (e.g., corn-soybean based diet compared to wheat bran, or barley) for enzyme action [8,11,12]. Therefore, these intriguing findings suggest that the compounds generated by enzyme digestion may have additional beneficial effects on intestinal microbiota, immune system activation, and antioxidant status, ultimately enhancing animal growth.

Moreover, it has been reported that the matrix of energy and P of ingredients can be improved as a result of improved digestibility, and enhanced utilization of previously inaccessible nutrients upon the inclusion of the exogenous enzymes [3,13,14,15]. Although most studies have been conducted under controlled experimental conditions with generally favorable sanitary management, it is important to note that in commercial farms, where environmental, immunological, and management challenges are more intense and frequent, the energy released by enzyme action may not necessarily translate into better animal growth [16]. This observation could potentially explain the lack of performance enhancement observed in cases where non-ruminant animals showed greater nutrient digestibility or improved intestinal health with enzyme supplementation [17,18,19,20].

On the other hand, the positive relationship between exogenous enzymes and intestinal benefits have been observed for at least a decade of research [6,21,22,23,24,25]. Thus, as part of the improvement in growth performance can be attributed to the beneficial effects of exogenous enzymes on intestinal health, their supplementation in diets of animals under poor sanitary conditions may prove to be superior [26,27]. However, only a limited number of publications have specifically investigated the effects of exogenous enzyme supplementation on intestinal health and performance under poor sanitary conditions. Therefore, this current review aimed to clarify the mechanisms through which exogenous enzymes (carbohydrases and phytase) influence intestinal health, as well as their potential effects on the nutrients and energy matrix in diets fed to poultry and pig reared under sanitary challenge.

## 2. Exogenous Enzymes and Their Functional Mechanisms of Action

Exogenous enzymes include all enzymes that are supplemented in the diet to break down the complex structure of cellular components by hydrolysis [12,28]. Xylanase, β-mannanase, β-glucanase, α-amylase, and phytase account for the majority of enzymes used nowadays (Figure 1). These enzymes may act on a variety of antinutritional factors such as non-starch polysaccharides, resistant starch, and phytate.

### 2.1. Carbohydrases

Carbohydrases include all enzymes that catalyze a reduction in the molecular weight of polymeric carbohydrates. Dietary carbohydrate fractions can impact the intestinal microbiota and absorptive function, and immune response in non-ruminant animals [29]. Cereal fibers contain several NSP components, such as arabinoxylans, cellulose and β-glucans, and resistant starch (RS), which can have an impact on overall health [2,30].

Soluble NSP (e.g., arabinoxylans and β-glucans) can have undesirable effects, such as increasing digesta viscosity and intestinal permeability [9,10,12], reducing beneficial microbiota, and promoting the growth of pathogenic bacteria in the intestine [31]. Consequently, viscous NSP impairs nutrient digestibility (e.g., reducing emulsification of dietary lipids, and altering short-chain fatty acids production), and causing changes in the nutrient interaction with the intestinal brush border [32]. Allied to this, the insoluble NSP forms a protective barrier, which reduces the usage of nutrients from the diet because it acts as a “cage effect” for cellular content [33].

In addition, NSP can directly interact with components of the immune system. Binding of NSP to immune cell surface receptors triggers several cellular and molecular pathways, leading to immune activation [34]. This immune stimulation process can increase the maintenance energy by 23% because the immune system requires substantial amounts of nutrients (e.g., amino acids), and energy that otherwise could be used for the animal’s growth (Figure 2) [35]. Recent studies suggest that the hydrolysis of polysaccharides by enzymes into oligosaccharides, acting as “prebiotics”, can beneficially modulate the intestinal microbiota [21,31]. This modulation results in improved intestinal health by increasing the production of short-chain fatty acids (SCFA) and reducing immune activation [11,21,36].

In this section, we will describe the functions of key carbohydrases (xylanase, β-mannanase, β-glucanase, and α-amylase) commonly used in feeds for non-ruminant animals to improve intestinal health and growth performance.

#### 2.1.1. Xylanase

Xylanases belong to six different glycosidic hydrolase (GH) families (GH5, GH7, GH8, GH10, GH11, and GH43), which exhibit different substrate specificities [34]. They are produced by several microorganisms (e.g., yeast, fungi, and bacteria), and commercially available xylanases are mainly derived from bacteria or filamentous fungi [37]. Arabinose and xylose are found as constituents of arabinoxylans (AX) in dietary ingredients [34]. The most abundant non-cellulosic polysaccharides in a corn-soybean meal-DDGS diet, and particularly in cereals, are AX.

Xylanase likely hydrolyzes the β-1,4-glycosidic bonds of the AX main chain into smaller fragments, such as xylo-oligosaccharides (XOS) or arabinoxylo-oligosaccharides (AXOS) [28]. Usually, the improved performance observed with exogenous xylanase has been attributed mainly to reduced viscosity and increased nutrient digestibility [38]. However, in addition to enhancing nutrient availability, xylanase-mediated AX hydrolysis can improve intestinal morphology [9,39], enhance intestinal barrier integrity [10,22,40], regulate immune function [10,11], and mitigate localized and systemic oxidative stress [9,40]. These mechanisms may help explain the improvements observed in animal growth.

The positive effects of xylanase supplementation on intestinal health result from the modulation of the microbiota in the ileum [41], cecum [42,43], and colon [44]. The most plausible mechanism for microbiome modulation is the “prebiotic” or “symbiotic” effect of XOS and AXOS released through AX hydrolysis [45,46]. These smaller fragments stimulate the proliferation of microbiota capable of efficiently degrading AX and are generally recognized as beneficial [41,47,48].

In addition to its ability to reduce pathogenic bacteria colonization in the intestine [31,41,42], which is associated with immune system activation, xylanase supplementation provides two additional mechanisms for immune function regulation. Firstly, xylanase can hydrolyze the polysaccharide AX, which can be recognized by pattern recognition receptors (such as toll-like receptors; TLR) on immune cell surfaces and can trigger the production of various cytokines and chemokines, thus influencing immune responses [34]. Secondly, xylanase supplementation can increase SCFA production [22,39]. Short-chain fatty acids, particularly butyrate, play a crucial role in energy supply, cell proliferation, and the regulation of immune responses [49,50]. Therefore, this increase in SCFA production can contribute to immune inactivation and overall immune system balance in pigs [10,11], and poultry [42].

The production of free radicals within the body can lead to damage to DNA or cell structures, thus affecting the stability of intestinal barrier function [51]. Recent studies have reported that exogenous xylanase improves the antioxidant status in pigs [9,10,40], and broilers [42,52], in both in vivo and in vitro trials. However, the specific mechanism by which xylanase decreases oxidative stress and enhances antioxidant capacity remains unclear. Petry et al. [40] suggested that this potential mechanism could involve the improvement of the bioavailability of phenolic compounds present within the AX structure, which can act as antioxidants.

These improvements in the immune and antioxidant status of non-ruminant animals fed exogenous xylanase could potentially explain the enhancements observed in the intestinal morphology and integrity [10,22,39,40,53]. Therefore, the improvement in the performance of animals fed xylanase in the diet goes beyond the utilization of “encapsulated” nutrients, but rather through the enhancement of overall intestinal health.

#### 2.1.2. β-Mannanase

β-Mannanases are endohydrolase enzymes that cleave randomly within the 1,4-β-d mannan chain of glucomannans, and galactomannans [54]. Mannan is the second most abundant hemicellulosic polysaccharide found in cereals and their by-products, following AX [55]. Similar to other dietary NSP fractions, β-mannans are not adequately degraded by digestive enzymes in the upper gastrointestinal tract of non-ruminant animals [56]. Thus, soluble β-mannans have been shown to increase the viscosity of intestinal contents, leading to reduced absorption of nutrients such as glucose and lipids in pigs [15,57], and poultry [58,59].

In vitro studies have shown that the β-mannanase enzyme hydrolyzes the β-mannan’s backbone, releasing short β-1,4-mannoligosaccharides (β-1-4-MOS) [60,61,62]. In addition, they showed that these shorter β-1-4-MOS promote lactic acid production and exhibit inhibitory effects on enteropathogenic bacteria (e.g., *Escherichia coli* and *Salmonella* spp.) in both monoculture and co-culture fermentations [60,61,62]. These findings suggest that the hydrolysis products of beta-mannans can be selectively used by beneficial intestinal microorganisms such as *Bifidobacteria* and *Lactobacilli*, which are positively associated with intestinal health [63].

Consistent with these findings, β-mannanase supplementation has been observed to impact the composition of the intestinal microbiota, increasing the abundance of intestinal-health-associated microbiota such as *Lactobacillus*, *Ruminococcaceae*, and *Akkermansia*, whereas reducing bacteria associated with intestinal disorders such as *Salmonella* spp. and *E. coli* [64]. In summary, these studies highlight the potential benefits of β-mannanase supplementation in influencing intestinal microbiota abundance and reducing the prevalence of enteropathogenic bacteria in non-ruminant animals.

In addition to serving as a fermentative substrate for the growth of pathogenic bacteria in the gastrointestinal tract, β-mannans share similarities with carbohydrate fractions found in the cell walls of microorganisms, making them recognizable by cell membrane mannose receptors [65]. As a result, β-mannans can be identified by the host’s immune system as pathogen-associated molecular patterns through various mannose receptors on the gastrointestinal tract’s cell surface [63]. This recognition leads to intestinal inflammation, oxidative damage, villus atrophy, and poor nutrient utilization [22,23,35].

In this context, a positive correlation has been observed between the concentration of acute-phase proteins (APP) in the blood and dietary β-mannans [35,65]. Additionally, native locust bean galactomannan was found to stimulate the production of tumor necrosis factor alpha (TNF-α) and β-hexosaminidase secretion in cells [66]. However, these effects were reversed when the native locust bean galactomannan was hydrolyzed by β-mannanase [66].

Therefore, the hydrolysis of intact native β-mannans by β-mannanase results in the formation of β-1-4-MOS fragments that can no longer be recognized by TLRs. This prevents hyperactivation of the immune system [63,65], which can improve villus integrity and promote better weight gain and feed efficiency. Consistent with this, pigs fed diets supplemented with β-mannanase showed reduced serum haptoglobin and IL-1α concentrations [35]. Another study demonstrated that the supplementation of β-mannanase reduced serum APP in birds challenged with *Eimeria* [67]. These findings suggest that β-mannanase has the potential to mitigate the activation of the immune system, allowing greater energy and nutrient availability for tissue deposition. However, further investigations are required to achieve a better understanding of the specific conditions under which these benefits are most pronounced, because no changes were observed in serum APP, such as haptoglobin and C-reactive protein in nursery pigs fed β-mannanase in their diet [68,69].

In general, the action of β-mannanase can lead to positive modulation of the microbiota, decreased activation of the immune system, and increased production of SCFA in the gastrointestinal tract [22,35,68]. These combined effects contribute to better intestinal morphology and improved animal growth performance [23,57].

#### 2.1.3. β-Glucanase

Extensive research has been conducted on various aspects of dietary fiber, with particular attention given to AX. However, β-glucan is another significant cell wall component, with higher concentration in barley and oats grains, followed by rye and wheat [70]. Compared to poultry research, studies involving pigs have been more extensive, as pigs are often used as models to assess the potential effects of β-glucan on human health [71,72].

β-Glucans are polymers consisting of D-glucose building blocks linked by β(1→3), β(1→6), or β(1→4) linkages [73]. They can be found in various feed components, including cereals, yeast, and mushrooms. Regarding β-glucans from cereals (e.g., oats and barley), they consist mainly of linear β(1→3) and β(1→4) linked glucose polysaccharides, connected by two or three consecutive β-(1-4) bonds, and separated by a single β-(1-3) bond [73]. In addition, the content and structure of β-glucan can vary within a specific cereal grain due to plant genetic, and environmental factors [56].

Although it is reasonable to consider β-glucanase as a suitable option for mitigating the negative effects of β-glucan found in barley and oat grain cell walls, most research studies have focused on evaluating the combined dietary supplementation of β-glucanase and xylanase. Considering that these grains also contain AX, the xylanase supplementation can enhance the effectiveness of β-glucanase, thus improving the feeding value of barley and oats [70]. Combined supplementation of β-glucanase and xylanase has been shown to improve weight gain and feed efficiency, as well as greater digestibility of energy, dry matter, and crude protein in poultry and pig, as evidenced by decreased digesta viscosity and increased NSP digestibility [18,31,74,75,76].

In addition to increasing the digestibility of energy and crude protein, Duarte et al. [31] observed that the supplementation of β-glucanase in diets containing xylanase attenuated the immune response by reducing plasma IL-6 and TNF-α concentrations in piglets. This reduction led to an increase in jejunal villus height, thus explaining the improved average daily gain (ADG). In this context, β-glucans can directly affect the immune system by stimulating dectin-1 [77]. The activation of dectin-1, in turn, stimulates the production of pro-inflammatory cytokines, effectively triggering the immune response [77,78]. Therefore, β-glucanase can reduce the activation of the immune system by preventing the stimulation of dectin-1 through the breakdown of β-glucan into smaller molecules (e.g., oligosaccharides), which results in improvements in villus length [23].

Another possible explanation for the attenuation of the inflammatory response is the modulation of the microbiota caused by the release of these oligosaccharides by β-glucanase. This modulation stimulates the growth of potential beneficial bacteria (e.g., *Faecalibacterium prausnitzii*) and reduces potential harmful bacteria (e.g., *Campylobacteraceae*, and *Helicobacter rappini*) in the intestine [31]. The hydrolysis of NSP releases oligosaccharides, increasing the fermentability of the dietary fiber by microbes and production of SCFA along the intestine which have been shown to inhibit inflammation within the intestine of pigs [45,46].

The same principle could be used to explain the beneficial effects of β-glucanase in energy metabolism, intestinal cell proliferation, and immune response [23,70]. However, recent studies have reported that dietary supplementation with β-glucanase reduces SCFA production in cecal [79] and colonic digesta [18] in poultry and pig, respectively, while still improving ADG and feed efficiency. Therefore, further research is needed to elucidate the effect of β-glucanase on SCFA production in the digesta.

In summary, these studies suggest that beneficial modulation of the microbiota appears to be the main mechanism by which β-glucanase reduces inflammation and improves intestinal morphology.

#### 2.1.4. α-Amylase

Corn, being the dominant starch source, is widely utilized in pig and poultry diets, and is provided as the main feed ingredient in quantitative terms to meet energy requirements [80]. Starch, a complex carbohydrate, constitutes two main components (amylose and amylopectin), with distinct structural traits [81]. Amylose is mainly composed of linear α(1-4)-linked anhydroglucose units, whereas amylopectin is highly branched due to additional α(1-6)-linkages [81].

Unlike NPS, starch can be digested by endogenous amylase in the animals’ intestines [82]. However, the post-weaning period in piglets and the early days of life in chicks are marked by a limited production of endogenous amylase, highlighting the importance of supplementing exogenous amylase in their diets to optimize the efficiency of starch digestion [80,83]. Furthermore, considering the substantial amount of starch fed to animals during the finishing phase, ranging from 40% to 60%, the existing endogenous amylase activity may be insufficient, further emphasizing the need to supplement exogenous amylase in their diets [12,84].

In this context, studies have documented the positive results of supplementation with exogenous α-amylase. These studies have highlighted improvements in nutrient and energy digestibility, as well as enhanced growth performance in broilers [12,85,86,87], and in pigs fed corn-soybean-based diets [88]. An increase in starch digestibility is accompanied by a greater release of other nutrients for endogenous enzymatic digestion because the starch granules are incorporated into a matrix that also contains protein and lipids [89]. Thus, a-amylase supplementation improves nutrient and energy digestibility, as demonstrated in broilers [90] and pigs [91].

However, studies evaluating the possible use of exogenous α-amylase on intestinal morphology, immune response, and microbiota are scarce. Nevertheless, Aderibigbe et al. [12] reported that broiler chickens fed diets containing exogenous α-amylase exhibited increased jejunal villus height. Similarly, Córdova-Noboa et al. [86] observed that dietary α-amylase supplementation led to an increase in the relative length of the small intestine in broilers. These findings indicate that the effects of α-amylase go beyond starch degradation, potentially contributing to improving intestinal health and subsequently enhancing growth performance. However, the mechanisms that explain how α-amylase can improve intestinal development are unclear.

Jiang et al. [92] observed a decline in pancreatic α-amylase activity ranging from 9% to 33% as exogenous α-amylase levels increased from a 250 mg/kg to a 2250 mg/kg diet. In fact, exogenous enzymes supplementation can be antagonistic to the secretion of some endogenous enzymes [90]. By reducing the need for pancreatic amylase synthesis and reducing intestinal and pancreatic mass [80,93], it is possible to save energy and amino acids, which can be used for enterocyte cell proliferation or other body functions, explaining the improvements observed in intestinal morphology.

Furthermore, the supplementation of exogenous α-amylase modulating the intestinal microbiota is due to the ability of α-amylase supplementation to act upon undigested starch fractions, which consist of resistant starch known for its “prebiotic” properties [90,94]. However, to the best of our knowledge, the impact of exogenous α-amylase on the microbiota in poultry or pigs has not yet been published.

### 2.2. Phytase

Phosphorus is an essential mineral for energy metabolism, nucleic acid synthesis, and the structure of cell membranes. In addition, the most important function of P is bone formation and mineralization, where it also serves as a reserve to be mobilized to play roles in almost all metabolic processes [95]. However, the P present in plant feedstuffs, commonly known as phytic P, has long been considered unavailable for non-ruminants [96].

In most grain-based ingredients used in pigs and poultry diets, phytic acid P can account for up to 80% of the total P [97]. Phytic acid (myo-inositol-1,2,3,4,5,6-hexakis [dihydrogen (phosphate)], C_6_H_18_P_6_O_24_) can form complexes with other minerals within the gastrointestinal tract of animals, making them unavailable [96,98]. Moreover, it can bind to amino acids, proteins, and enzymes (e.g., trypsin and α-amylase), inhibiting their activity and affecting protein and carbohydrate digestibility [99,100].

Phytic P is partially utilized in non-ruminant animals due to low endogenous phytase activity in the small intestine, which is insufficient to digest phytate and release P for absorption [101]. Consequently, high levels of inorganic P sources are added to diets to meet specific nutritional requirements [102,103]. Thus, nutritional strategies have been evaluated to increase the P bioavailability from phytic P.

Phytases (myo-inositol hexakiphosphate phosphohydrolase) are responsible for catalyzing the hydrolysis and release of P from phytic acid present in plant-based feedstuffs [102,104]. In this sense, this enzyme is supplemented in non-ruminant diets to increase the digestibility of phytic P, leading to less dependence on inorganic P and minimizing the excretion of this mineral into the environment [4,105].

Furthermore, the hydrolysis of phytate molecules increases the availability of other minerals (e.g., Ca, Na, K, Mg, and Zn), amino acids (e.g., Lys, Cys, Thr, Val, Ile, Leu, Thr, His, Arg, and Phe), and energy, improving the utilization of dietary nutrients for animal growth performance (Figure 3) [4,106,107]. The supply of inositol, a product of phytate degradation, has also received more attention as phytase supplementation becomes more common [108]. This extra release of nutrients is the “so-called” extra-phosphoric effect of phytase, which is responsible for a portion of the increase in growth performance associated with phytase supplementation [103,109,110]. In addition, these nutrients have been found to positively influence the microbiota, immune response, and antioxidant status in non-ruminant animals [111,112,113], which is reflected in improved intestinal morphology and integrity [114,115,116]. However, the precise mechanisms by which phytase improves overall intestinal health are not fully understood.

Reducing undigested phytic acid in the intestine can have significant effects on the intestinal environment. One notable effect is the reduction of the digesta pH, thus creating a favorable environment for the growth of beneficial bacteria and inhibiting the proliferation of harmful ones [116,117]. In addition, phytase can potentially exert a positive impact on the microbiota through a mechanism that involves inducing intestinal alkaline phosphatase activity [118]. In addition to its role in dephosphorylating inositol monophosphate, intestinal alkaline phosphatase performs several functions, including dephosphorylating bacterial lipopolysaccharide (LPS) and preventing bacterial transepithelial passage [118,119,120].

In agreement, Moita et al. [116] found that the inclusion of a 2000 FTU/kg diet tended to decrease harmful bacteria such as *Pelomonas*, *Helicobacter*, and *Pseudomonas*, whereas increasing beneficial bacteria such as *Lactobacillus* in broilers. Similarly, Ptak et al. [121] reported an increase in *Lactobacillus* populations and a decrease in *Streptococcus* abundance in the ileum of broiler chickens fed 5000 FTU/kg. In addition, Nari et al. [24] observed reduced bacterial counts of *E. coli* and *Clostridium* spp. in the ileum of broilers fed 500 FTU/kg.

In pigs, Liu et al. [25] observed decreases in *Tenericutes* and *Spirochaetes* at the phylum level in the cecum of pigs fed 500 U/kg. These phyla include a diverse range of species, which can have harmful effects in animals [122]. Furthermore, Moita and Kim [123] found a reduced abundance of the Prevotellaceae family in the jejunum of nursery piglets fed 2000 FTU/kg, which aligns with the findings of Liu et al. [25]. The Prevotellaceae family is known for its diversity, with some members playing a crucial role in the fermentation of dietary fiber within the intestine [124]. This suggests that phytase supplementation could potentially have an impact on SCFA production. However, the overall effects of phytase on the intestinal microbiota, mainly in pigs, remain unclear and need further investigation.

Specific lower phosphorylated inositol phosphates, such as myo-inositol triphosphates (InsP3) and myo-inositol tetraphosphates (InsP4), have been shown to play a significant role in cell signal transduction, cell function regulation, cell growth and differentiation [125,126]. The complete hydrolysis of phytate results in the production of myo-inositol, which can be absorbed and detected in both portal and peripheral blood [127,128]. Myoinositol may increase insulin sensitivity and also may promote insulin secretion from pancreatic β cells [128]. Thus, myo-inositol might have an insulin-mimicking effect in the stimulation of glucose uptake into tissues [126,127], which ultimately leads to improved growth performance. Studies have reported that dietary supplementation with phytase can raise the blood concentration of myo-inositol in broiler chickens [129,130], and pigs [108,128,131]. Therefore, this increase in myo-inositol concentrations may contribute to a reduced immune response and improved antioxidant status in these animals.

In line with this, Zhang et al. [111] demonstrated a decrease in IL-1β and TNF-α concentrations in the jejunal mucosa of pigs fed a diet containing 1000 U/kg. In addition, these animals exhibited higher glutathione peroxidase (GSH-Px) and catalase (CAT) activities in the duodenum and ileum, respectively. Similarly, Ren et al. [114] reported a reduction in the plasma malondialdehyde (MDA) concentration in nursery pigs fed 500 FTU/kg. However, Moita and Kim [123] observed no effects on immune response and antioxidant status in nursery piglets fed 2000 FTU/kg.

In a study conducted with broiler chickens, Adedokun and Adeola [112] demonstrated that the supplementation of a 5000 FTU/kg diet resulted in a reduction of IL-6 mRNA expression in the jejunal mucosa. It also decreased IL-1β mRNA expression in animals that were not challenged with coccidial vaccines. In a study conducted by Khodambashi Emami et al. [132], it was found that supplementation with 500 FTU/kg increased the concentrations of total Ig, IgM, and IgG in broilers on the primary (d21) and secondary (d28) antibody response against sheep red blood cells. These improvements in the immune response may contribute to the antioxidant status of broilers, as indicated by Derakhshan et al. [113], who observed that phytase supplementation increased the antioxidant activities of GSH-Px, CAT, superoxide dismutase (SOD), and total antioxidant capacity. In addition, they reported a reduction in serum MDA concentration as a biomarker of oxidative stress.

Considering all the extra-phosphoric effects resulting from phytase supplementation in non-ruminant animal diets, such as the beneficial modulation of the microbiota, reduced activation of the immune system, and improved antioxidant status, it is plausible that there are enhancements in intestinal morphology and barrier function. This has been demonstrated in studies conducted with broiler chickens [24,115,116,132], and pigs [25,114,131], which suggests that improvements in growth performance cannot only be attributed to increased nutrient availability, but also to overall improved intestinal health.

## 3. The Role of Dietary Exogenous Enzymes in Poultry and Pig under Challenging Conditions

The negative impact of an immune challenge on animal growth has been widely documented [133,134]. Furthermore, pro-inflammatory cytokines regulate an immune response characterized by fever, APP production, and leukocyte proliferation, leading to oxidative stress and apoptosis in intestinal epithelial cells. These processes require extra energy and amino acids to sustain the immune system’s requirements [135,136]. Consequently, the presence of an immune challenge has the potential to redirect energy and nutrients from anabolic processes such as muscle growth, causing detrimental effects on production costs [135,137,138]. Supporting this notion, Huntley et al. [35] revealed that an innate immune challenge in weaned piglets resulted in a significant increase in pro-inflammatory cytokine concentrations, leading to a remarkable 23.3% elevation in metabolizable energy (ME) required for maintenance. Consequently, this metabolic change correlated with an 18.3% decrease in ADG during the LPS challenge. Similarly, Dunaway and Adedokun [139] observed a 20.0% reduction in apparent ME corrected for nitrogen retention (AMEn) in broiler chickens seven days after administration of an oral coccidia vaccine.

As discussed above, the carbohydrases supplementation such as xylanase, β-mannanase, β-glucanase, and α-amylase, together with phytase, not only increases the nutrient availability in the gastrointestinal tract, but also promotes positive effects on intestinal health. These effects encompass attenuating inflammatory responses, reducing oxidative stress, and improving intestinal morphology and integrity. However, it is important to note that most of these studies were conducted under controlled experimental conditions, typically characterized by favorable sanitary handling.

In contrast, commercial production systems with poor sanitary conditions have a greater risk of bacterial infection, which can have a negative impact on growth rate and feed efficiency [26,140]. Consequently, the effectiveness of exogenous enzyme supplementation may be affected in circumstances where hygiene conditions are suboptimal and stress is frequent [26,27]. Over the last year, there has been an increased interest in the investigation of the effects of carbohydrases and phytase on sanitary-challenged poultry and pigs (Table 1). Based on the compiled data (Figure 4), when enzyme was supplemented animals become more efficient at converting energy into weight gain. For non-challenged or challenged animals, the conversion of ME intake into BW gain showed 1.9% and 3.4% improvements associated with enzyme supplementation.

Supporting this hypothesis, Duarte et al. [45] demonstrated that a feed additive based on xylanase and probiotic was effective in reducing the jejunal mucosa levels of IL-6, TNF-α, and protein carbonyl regardless of the *E. coli* challenge. Moreover, challenged and non-challenged pigs fed with the additive presented improved growth performance and reduced diarrhea than those without additive. In an experiment conducted in situ, Kiarie et al. [155] reported that carbohydrase hydrolysis products obtained from a soybean meal had positive effects against enterotoxigenic *E. coli* infection in piglets. Cho et al. [26] demonstrated that the supplementation of a feed additive based on β-1,4-endo-xylanase and xylo-oligosaccharides had limited effects on the growth performance and intestinal health of weaned piglets under good sanitary conditions. However, under poor sanitary conditions, the additive supplementation could mitigate the immune challenge, as demonstrated by the improved growth performance and lower inflammatory response.

Similarly, Song et al. [153] reported that the supplementation with β-1,4-endo-xylanase and xylo-oligosaccharides exhibited performance-enhancing effects, with a reduction in the blood concentration of pro-inflammatory cytokines (IL-6 and TNF-α), increased villus height, and decreased crypt depth in weaned piglets. Notably, these beneficial effects were particularly pronounced in animals challenged with *E. coli*. In another study, enzyme blend (xylanase, β-glucanase, and pectinase) supplementation improved the ADG of piglets during the pre- and post-F18-ETEC challenge period, which is partly explained by an increase in intestinal barrier integrity markers (occludin and zonula occludens-1 mRNA) and a reduction in inflammation markers (TNF-α mRNA and serum haptoglobin) [152].

In a study involving broiler chickens, Dersjant-Li et al. [138] demonstrated that the supplementation of a feed additive based on an enzyme blend (xylanase, α-amylase, and protease) and *Bacillus* spp. resulted in improved growth performance facing a coccidia challenge. Moreover, the feed additive demonstrated its ability to mitigate the inflammatory response induced by the coccidia challenge. This was evident in reduced plasma APP concentrations (e.g., hemopexin and α-1-acid glycoprotein), a decreased number of intraepithelial lymphocytes, a thinner lamina propria, and reduced concentrations of inflammatory markers such as duodenal IL-6 on day 21 compared to the challenged control group. Similarly, in a study conducted by Lee et al. [145], it was observed that broiler chickens challenged with *C. perfringens* exhibited elevated jejunal and ileal lesion scores, increased villus atrophy, and higher plasma TNF-α and endotoxin concentrations. However, xylanase and xylo-oligosaccharides supplementation demonstrated its capability to mitigate these detrimental effects and restore intestinal health in birds. In addition, broilers challenged with *C. perfringens* had improved growth performance when fed diets containing carbohydrases [143,156].

Regarding phytase supplementation under poor sanitary conditions, Moran et al. [157] reported a quadratic response in the overall gain-to-feed ratio of weaned piglets fed increasing doses of phytase, reaching an optimum dose with 2500 FTU/kg. In addition, the authors observed enhanced feces firmness in piglets fed 2500 FTU/kg compared to those fed 600 FTU/kg, indicating improved intestinal health. Consistent with these findings, Moran et al. [108] also observed better growth performance in weaned piglets fed 2500 FTU/kg reared under poor sanitary conditions.

However, whereas there are studies showing the benefits of these exogenous enzymes on intestinal health and performance under sanitary challenging conditions, others have reported little or no effect of enzyme supplementation [16,35,139,146]. In this way, further studies are needed to elucidate the effects of carbohydrase and phytase in alleviating overall intestinal damage in non-ruminant animals subjected to sanitary challenges.

Moreover, it is important to highlight that most of the studies discussed so far, which have investigated enzyme supplementation in pig and poultry diets under various sanitary challenging conditions, have not involved reducing the energy and/or nutrients density of the supplemented diets to understand whether, in fact, the nutritional matrix values should be altered.

In commercial practice, the use of enzymes is commonly associated with the incorporation of a nutritional matrix that considers the predicted effects. When a matrix is accurately applied to a feed enzyme, it can lead to significant savings in diet costs [158]. The main classes of enzymes are associated with specific matrices, including the following: carbohydrases use energy, amino acids, and protein matrices, whereas phytases employ matrices of P, Ca, Na, amino acids, protein, energy, and other minerals [158].

Regarding the energy matrix, studies have reported that the supplementation of carbohydrases (e.g., xylanase, β-mannanase, β-glucanase, and α-amylase) and phytase provides formulators with the opportunity to reduce the ME content of the diet by around 100 kcal/kg in diets to broiler chickens [159,160,161,162,163], and pigs [13,40,74,164]. Recently, de França et al. [165] reported that the supplementation of enzymes (xylanase, β-mannanase, β-glucanase, and phytase) in broiler diets makes it possible to reduce the energy density by a 200 kcal/kg diet without affecting growth performance and jejunal morphometry.

Although these studies were conducted under experimental conditions and controlled sanitary management, it is important to note that in commercial production systems, where activation of the immune system occurs more frequently, the additional energy released as a result of enzyme activity may not necessarily translate into better growth performance, since a portion of the energy and nutrients are allocated to support the immune response [16]. On the other hand, since exogenous enzymes can improve intestinal health, as already discussed in this review, their incorporation into commercial conditions has the potential to enhance energy utilization. As a result, there is a possibility that the energy matrix developed in research facilities may underestimate the true impact of these enzymes. Consequently, the energy matrix of the enzymes may be altered under poor sanitary conditions.

In this context, a study conducted by Nusairat and Wang [166] demonstrated that the supplementation of xylanase in diets with a 130 kcal/kg reduction in ME, at a level of up to a 15.0 XU/g diet, resulted in improved body weight gain and feed conversion ratio, enhanced energy digestibility, and reduced intestinal lesion scores in broilers reared under typical production conditions. Bello et al. [167] also demonstrated that the supplementation of 2.0 XU/g xylanase in diets formulated with a 75 kcal/kg reduction in ME maintained all growth performance variables of broilers reared in an environment with a high pathogenic load, per phase and cumulatively. Similarly, in a study conducted by Ennis et al. [168], who assessed used litter bedding obtained from commercial broiler houses, observed that broilers fed ME-reduced diets (−100 kcal/kg) supplemented with β-mannanase or xylanase exhibited growth performance results similar to those of birds fed a standard diet. Using a diet (based on corn, soybean meal, and wheat middling) formulated with lower energy and digestible amino acids compared to breeders’ recommendations, Dersjant-Li et al. [169] demonstrated that using a combination of multi-enzyme (xylanase, amylase, and protease) and strains of *Bacillus amyloliquefaciens* for broilers can result in improved feed efficiency and ADG, and a lower foot-pad lesion score under commercial environments.

These studies indicate that the energy matrix of carbohydrases can be effectively applied in broiler production under commercial conditions. However, the extent to which ME can be saved in such conditions remains unclear. In addition, published research on the assessment of carbohydrases in pig production at the farm level is limited.

Phytase allows formulators to reduce available P, total Ca, ME, and other nutrients in diets without compromising growth performance. This has been supported by studies which have shown that phytase supplementation in diets containing levels of available P, total Ca, and ME below the recommendations by approximately 0.15%, 0.16%, and 100 kcal/kg, respectively, can maintain growth performance and intestinal health in broilers [24,163,170,171], and pigs [123,131,164,172].

In the same context as carbohydrases, phytase matrices developed in research facilities can be altered under poor sanitary conditions due to constant immune challenges. However, recent studies [167,168,173] have indicated that phytase matrices can be successfully implemented in non-ruminant production under commercial conditions without any adverse effects on growth performance.

For instance, Dersjant-Li et al. [173] conducted a study involving broiler chickens, in which the matrix supply of a *Buttiauxella* phytase made it possible to reduce the levels of ME, total Ca, and available P by 67 kcal/kg, 0.17%, and 0.16%, respectively. Despite these reductions, the broilers maintained tibia ash, growth performance, slaughter yields, and carcass yields at levels equivalent to those observed in broilers fed a nutritionally adequate diet when tested in a commercial environment. Similarly, Bello et al. [167] observed that supplementation of 6-phytase to diets with reduced levels of ME (−75 kcal/kg), total Ca (−0.15%), and available P (−0.17%) could maintain or even improve the growth performance, bone mineralization and breaking strength of broilers compared to a nutritionally adequate diet. In addition, Ennis et al. [168] reported that the *E. coli*–derived 6-phytase supplementation at a level of 1500 FTU/kg to a diet with reduced total Ca and available P levels of 0.145%, and AME at 100 kcal/kg, clearly improved the broiler growth performance by a better feed conversion ratio of 4 and 3 points during the 28-day and 44-day periods, respectively. Phytase super dosing (1500 FTU/kg) also led to improved 45-day processing yields, increased tender yields relative to carcass weight, and showed a trend towards reducing fat pad weights.

In the case of pigs housed under commercial conditions, Cambra-López et al. [174] demonstrated that supplementation of a 1000 FTU/kg diet with total P levels of 13%, 24%, and 23% below the recommendations for weaned, grower, and finisher pigs, respectively, has the potential to improve Ca and P digestibility, digestible energy, growth performance, and bone mineralization. The authors also observed that the effectiveness of 3-phytase decreased during the finisher phase, suggesting that the importance of this enzyme in promoting intestinal health, particularly in the susceptible early phases, could amplify its positive impact on pig growth performance. Similarly, Dersjant-Li et al. [175] observed that the supplementation of phytase from *Buttiauxella* sp. (500 or 1000 FTU/kg) in diets without inorganic phosphate and with reduced Ca (−0.13%) and ME (−35.8 kcal/kg) levels could maintain the growth performance and carcass traits of commercial pigs from 12 kg BW until slaughter.

It is important to note that all of these studies were conducted in commercial facilities or designed to simulate such conditions. Therefore, phytase matrices for energy and minerals seem to be effective in enhancing the growth performance of broilers reared under poor sanitary conditions. However, more studies are needed to fully understand the extent of their effects.

These findings collectively suggest that the energy and nutrient matrix released by the action of carbohydrases and phytase can be applied for animals on commercial farms, but the extent to which the matrix can be valued under challenging conditions needs further investigation.

## 4. Conclusions and Future Directions

As highlighted in this review, supplementation with exogenous enzymes offers benefits that go beyond the utilization of “encapsulated” nutrients. It positively modulates the intestinal microbiota, attenuates the activation of the immune system, and improves the antioxidant status of animals. Despite the demonstrated ability of enzymes to save energy and nutrients, their responses under challenging conditions have been poorly documented. Activation of the immune system alters nutrient and energy partitioning, potentially affecting the value of developed matrices in commercial farms. Interestingly, under sanitary challenges, supplementation of carbohydrases and phytase can be implemented using the energy and nutritional valorization matrices. This suggests that enzymes, with their properties that promote overall intestinal health, expedite the recovery of intestinal function after sanitary challenges, while maintaining the improved utilization of the nutritional matrix.

Studies conducted under commercial conditions have shown that the use of matrices containing carbohydrases and phytase can maintain the growth performance and general health of broiler chickens and pigs. However, these studies predominantly focused on assessing a single level of reduction in energy and/or available P and total Ca, limiting our ability to quantify potential energy and nutrient savings in the diet. Therefore, further studies should be conducted with a specific focus on determining the extent of energy and nutrients savings, as well as seeking to understand the effects of supplemented enzymes alone, and in complexes or blends.

## Figures and Tables

**Figure 1 animals-14-00226-f001:**
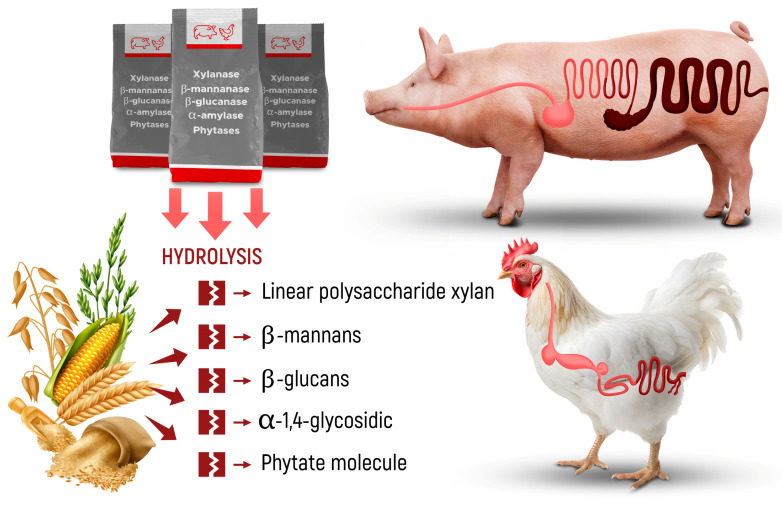
Most used exogenous enzymes in non-ruminant diets and their substrate of action.

**Figure 2 animals-14-00226-f002:**
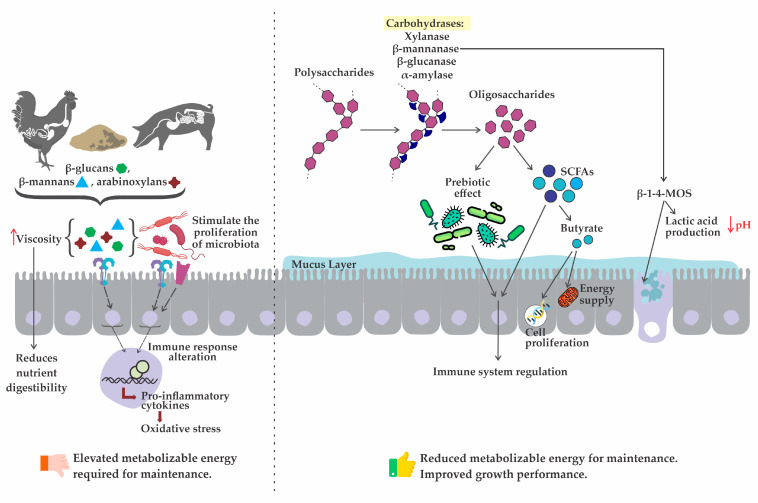
Harmful effects of non-starch polysaccharides and the beneficial effects of exogenous carbohydrases supplementation in non-ruminant diets.

**Figure 3 animals-14-00226-f003:**
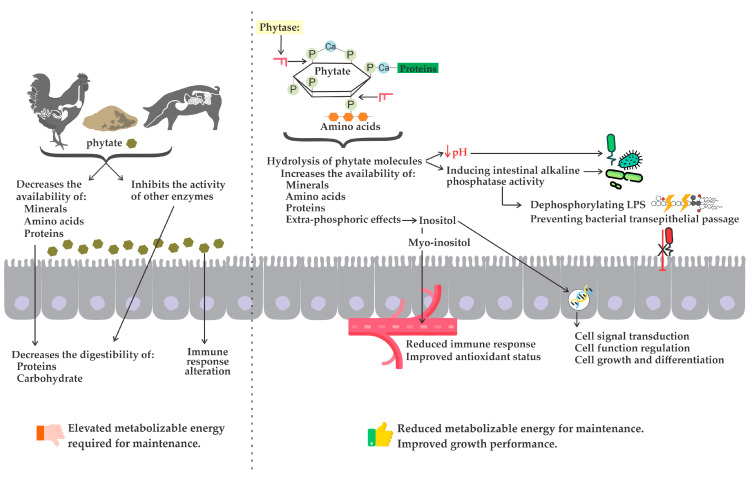
Harmful effects of a phytate molecule and the beneficial effects of phytase supplementation in non-ruminant diets.

**Figure 4 animals-14-00226-f004:**
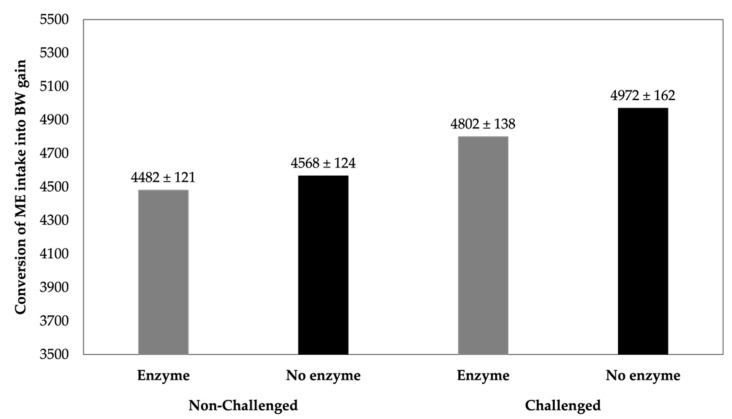
Estimated calorie conversion (kcal consumed per kg of BW gain) for sanitary challenged and non-challenged poultry and pigs. The described effects were based on treatments supplemented with carbohydrase or phytase in an isolated or blend form in comparison with treatments containing no enzyme. Data based on studies displayed in Table 1.

**Table 1 animals-14-00226-t001:** Studies evaluating the effects of carbohydrases and phytase on performance and/or intestinal health of broilers and pigs using a sanitary challenge model.

References	Species	Diet Type	Evaluation Period	Enzymes	Challenge Type
[112]	Broiler	Corn/SBM	49 to 55 d	Phytase 1000 FTU	Coccidial vaccine
[112]	Broiler	Corn/SBM	49 to 55 d	Phytase 5000 FTU	Coccidial vaccine
[141]	Broiler	Wheat/Barley/DDGS	1 to 35 d	Xylanase + protease	Necrotic enteritis
[142]	Broiler	Wheat/Corn/Barley/SBM	14 to 21 d	Xylanase	Coccidiosis vaccine
[142]	Broiler	Wheat/Corn/Barley/SBM	14 to 21 d	Xylanase + β-glucanase	Coccidiosis vaccine
[138]	Broiler	Corn/Wheat/Rye/SBM	1 to 21 d	Xylanase + amylase + protease + *Bacillus* spp.	Oral coccidia
[143]	Broiler	Corn/SBM	1 to 40 d	Cellulase + pectinase + xylanase + glucanase + mannanase + galactanase	Necrotic enteritis
[143]	Broiler	Wheat/SBM	1 to 40 d	Cellulase + pectinase + xylanase + glucanase + mannanase + galactanase	Necrotic enteritis
[144]	Broiler	Wheat/SBM	1 to 21 d	Xylanase	Necrotic enteritis
[144]	Broiler	Maize/SBM	1 to 21 d	Xylanase	Necrotic enteritis
[144]	Broiler	Wheat/SBM	1 to 21 d	β-Mannanase	Necrotic enteritis
[144]	Broiler	Maize/SBM	1 to 21 d	β-Mannanase	Necrotic enteritis
[145]	Broiler	Corn/Wheat/SBM	1 to30 d	Xylanase + xylo-oligosaccharides	Necrotic enteritis
[146]	Broiler	Corn/SBM	1 to 21 d	Xylanase + protease	Coccidiosis
[16]	Broiler	Corn/SBM	1 to 42 d	β-Mannanase	Coccidiosis
[147]	Broiler	Wheat/SBM	1 to 35 d	Xylanase + β-glucanase + β-Mannanase	Necrotic enteritis
[148]	Broiler	Corn/SBM (adequate Ca and P)	5 to 15 d	Phytase 600 FTU	Coccidiosis
[148]	Broiler	Corn/SBM (lower Ca and P)	5 to 15 d	Phytase 600 FTU	Coccidiosis
[149]	Broiler	Wheat/SBM	1 to 28 d	Phytase 5000 FTU	Necrotic enteritis
[150]	Broiler	Wheat/SBM (high Ca)	1 to 42 d	Phytase 1500 FTU	Necrotic enteritis
[150]	Broiler	Wheat/SBM (low Ca)	1 to 42 d	Phytase 1500 FTU	Necrotic enteritis
[151]	Pig	Barley/Wheat/SBM	21 to 35 d	Carbohydrases	*E. coli* (K88)
[26]	Pig	Wheat/Corn/SBM	21 to 63 d	Xylanase + xylo-oligosaccharides	Poor sanitary condition
[45]	Pig	Corn/SBM/DDGS	21 to 41 d	Xylanase + *Bacillus* sp.	*E. coli* (F18+)
[152]	Pig	Corn/SBM/DDGS	23 to 37 d	Xylanase + β-glucanase + pectinase	*E. coli* (F18+)
[152]	Pig	Corn/SBM/Sugar beet pulp	23 to 37 d	Xylanase + β-glucanase + pectinase	*E. coli* (F18+)
[153]	Pig	Corn/SBM	28 to 42 d	Xylanase + xylo-oligosaccharides	*E. coli*
[154]	Pig	Corn/Wheat/SBM	21 to 32 d	Cellulase + β-mannanase + galactanase + xylanase + β-glucanase + amylase + protease	*E. coli* (LPS)

## Data Availability

Publicly available datasets were analyzed in this study. This data can be found here: https://www.ncbi.nlm.nih.gov/ (accessed on 6 January 2024).

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
