# Peer review of "Carbohydrases and Phytase in Poultry and Pig Nutrition: A Review beyond the Nutrients and Energy Matrix"

_animals, 2024, doi:10.3390/ani14020226_

Round 1
Reviewer 1 Report
Comments and Suggestions for Authors
Overall, this is a nice review discussing non conventional roles such as immunoregulatory and healthy microbiota promoting roles of carbohydrases and phytases in poultry and pigs. I only have minor comments.
The simple summary and abstract are difficult to understand all together, and must be rewritten. Please see my comments about it below.
1. The Simple Summary does not sound simple. What's the main point you are trying to make? Please rephrase.
2. line 27-28: In abstract I don't understand what you meant by "their potential effects on the nutrients and energy matrix in diets fed to poultry and pig reared under challenging conditions", because it is not clear what are the challenging conditions, please rephrase or explain.
3. line 29: I think there's language error/inconsistency. "enzyme supplementation extends benefits by positively influencing", what do you mean by extend benefits, extend what benefits?
4. line 29-30: What does positively influencing microbiota mean? What does mitigating immune system mean? Terms like these are vague and redundant, please be more specific in terms of the effects, are specific microbiota species enriched, are any specific immune cell types suppressed or activated, what exactly is the point?
5. line 32: What is the challenging condition? Please be more specific.
6. line 33: What exactly is an acute challenge in the context of this review? Please explain.
7. lines 52-57 imply, phytase has a role in increasing bioavailability of P and increasing the matrix energy, but lines 67-68 suggests otherwise, thus contradicting whatever is written before. Please rectify or explain.
8. lines 78-81: I don't understand this sentence, please rephrase.
9. line 87: Challenging condition is a very vague term. As far as I understand, your review focuses on animals raised under non laboratory conditions. It does not necessarily mean its a challenging condition. What defines a challenge here? It is not clear.
10. line 88: Application of exogenous enzymes and their functional mechanisms, this is a very vague title. What does Functional mechanisms mean? Do you mean mechanisms of action, and what is application, applying somewhere or properties? Please rephrase the title and improve its clarity.
11. After reading section 3 it is clear that, the authors refer to "immunologically challenging conditions" when they infer challenging conditions. Please rephrase accordingly for clarity, and define it at the first instant it is discussed in this review.
Comments on the Quality of English Language
Only minor edits required.
Reviewer 2 Report
Comments and Suggestions for Authors
This is a well written manuscript concerning the potential effects and its mechanism of exogenous enzymes (carbohydrases and phytase) supplementation in poultry and pig diets. Unfortunately, it seems that the meanings of “matrix” and “matrices” are unintelligible for readers, and the content above certain enzyme seems to be repeated in the context. The discussion of the effects of exogenous enzymes on the excretion of organic matter, nitrogen, and phosphorus, which should be addressed in this paper as a complete scientific one in Part 2, since this topic of environmental protection arouses more attention in poultry and pig production nowadays.
Reviewer 3 Report
Comments and Suggestions for Authors
